# Digital health interventions for colorectal cancer screening uptake: A scoping review

Sujin Kim[1]*, Andrew J. Whipkey[2], Jihye Bae[3], Avinash Bhakta[2]

1 Division of Biomedical Informatics, Department of Internal Medicine, College of Medicine, University of Kentucky, Lexington, Kentucky, United States of America, 2 Department of Surgery, College of Medicine, University of Kentucky, Lexington, Kentucky, United States of America, 3 Department of Electrical and Computer Engineering, College of Engineering, University of Kentucky, Lexington, Kentucky, United States of America

* sujinkim@uky.edu

## Abstract

Digital health interventions (DHIs) are increasingly employed to improve colorectal cancer (CRC) screening uptake, yet comprehensive syntheses of their effectiveness across diverse contexts remain scarce. This scoping review examines how individual, contextual, technological, and timing-related factors shape CRC screening outcomes in DHI-based trials. Following PRISMA-ScR guidelines, we conducted a systematic search of PubMed, Google Scholar, and ClinicalTrials.gov from March 1 to April 20, 2024, identifying 4,523 records through databases and an additional 2,039 through backward citation tracking. After deduplication and screening, 51 studies were included and charted using the PICOT (Population, Intervention, Comparison, Outcome, and Timing) framework. Included studies spanned urban health systems, rural community clinics, and Federally Qualified Health Centers in the United States, Europe, Asia, and Australia, with intervention durations ranging from six weeks to ten years. Keyword co-occurrence mapping revealed four thematic domains: (1) patient-centered technology and adherence, (2) behavioral design and personalization, (3) clinical workflow and provider interaction, and (4) equity, disparities, and community engagement. Findings showed that tailored telephone outreach, mailed fecal immunochemical testing combined with navigation support, EMR-based automated reminders, and mobile applications delivering personalized education increased screening rates by 20.9% to 37.7% compared with conventional approaches. Hybrid models combining digital tools with human facilitation were particularly effective for underserved populations, including racial and ethnic minorities, rural communities, and individuals with limited health literacy. However, research gaps persist for younger adults at risk for early-onset CRC and for understanding the long-term sustainability and cost-effectiveness of digital interventions. Temporal aspects such as intervention timing, frequency, and duration were identified as important factors but were inconsistently reported. Future research should address

the Creative Commons CC0 public domain dedication.

**Data availability statement:** All relevant data are provided in the appendices. The articles reviewed in this scoping review are available through journal subscriptions, or in publicly accessible formats where applicable.

**Funding:** This publication was supported by the University of Kentucky College of Medicine Artificial Intelligence in Medicine Research Alliance (support to SK), and by the Center for Clinical and Translational Science, funded by the National Center for Research Resources and the National Center for Advancing Translational Sciences, National Institutes of Health, through Grant UL1TR001998 (support to SK). The funders had no role in study design, data collection and analysis, decision to publish, or preparation of the manuscript.

**Competing interests:** The authors have declared that no competing interests exist.

digital health literacy, implementation barriers, and long-term follow-up to support sustained CRC screening adherence through user-centered, scalable, and culturally responsive digital solutions.

---

## Author summary

Colorectal cancer (CRC) is one of the leading causes of cancer-related deaths, yet many people do not complete routine screening. We conducted this review to better understand how digital tools—like mobile apps, patient portals, automated messages, and telehealth—can help increase CRC screening, especially in populations that are often underserved or harder to reach. We reviewed 51 clinical studies conducted in various settings, including urban and rural clinics, and examined how technology was used to support patient education, reminders, and access to care. Our findings show that digital tools can significantly improve screening rates, especially when paired with personal support such as patient navigators or telephone follow-ups. However, we also found major gaps in research. For example, few studies focused on younger adults who are increasingly at risk for early-onset CRC, or on how long the benefits of digital interventions last. Many studies also lacked information about how well people understood or engaged with digital tools. We hope our review helps researchers and healthcare providers design better, more inclusive digital health programs to reduce disparities and improve CRC screening outcomes across different communities.

## Introduction

Colorectal cancer (CRC) is the third most common malignancy globally and remains a leading cause of cancer-related mortality, particularly in high-income countries [1]. The burden of CRC has been rising steadily in several parts of the world, with over 1.8 million new cases and nearly 900,000 deaths annually [2]. Despite advances in CRC screening protocols—such as colonoscopy, stool-based tests, and sigmoidoscopy—adherence to these life-saving guidelines remains alarmingly low in many regions [3,4]. Adherence is particularly challenging in low-income populations, where barriers such as limited access to healthcare services, lack of health literacy, and socioeconomic constraints further complicate screening efforts [5].

Timely screening can prevent up to 60% of CRC-related deaths [6,7], yet screening rates in many countries fall far short of the targets set by public health authorities [8]. Several strategies, including phone outreach programs, mailed reminders, and even financial incentives, have been deployed to boost CRC screening rates, with mixed success [9,10]. While these methods have proven somewhat effective, they often fail to reach vulnerable populations, such as those with limited health literacy or those residing in rural areas with insufficient healthcare access [11]. For instance, one large-scale program offering mailed fecal immunochemical test (FIT) kits to

low-income residents saw only modest increases in screening rates, indicating that more personalized and adaptable approaches are needed [12–14].

A growing body of evidence suggests that digital health interventions (DHIs) can address many of the barriers that traditional approaches have struggled to overcome. These technologies, which include mobile health (mHealth) apps, telehealth platforms, electronic medical records (EMR) systems, and patient portals, offer scalable and personalized solutions for CRC screening adherence. By leveraging these digital tools, healthcare providers can engage patients more effectively, particularly those in hard-to-reach populations, offering reminders, educational materials, and even remote consultations to reduce barriers such as misinformation, logistical challenges, and anxiety surrounding invasive procedures [15–18]. Research has shown that DHIs can significantly improve patient engagement and adherence by providing real-time support and tailoring interventions to individual patient needs. Additionally, these tools can increase access to care in underserved communities, a critical advantage for addressing disparities in CRC screening rates [19,20].

The COVID-19 pandemic further accelerated the adoption of telehealth and other digital services, making them an integral part of routine clinical care, including cancer screening. This shift toward digital platforms highlights the potential of DHIs to enhance screening uptake and ensure continuity of care in both urban and rural settings, despite external disruptions [21]. As a result, digital interventions are emerging as a key component in modernizing cancer prevention strategies, offering new opportunities to close the gap in CRC screening adherence. In light of these advances, this scoping review seeks to map current evidence on the use of DHIs to increase CRC screening uptake, characterize intervention strategies and populations targeted, and identify key research gaps. By comparing traditional screening methods with newer digital interventions, we aim to identify research gaps and inform future strategies for integrating these technologies into routine clinical practice. Specifically, this review addresses the following research questions: (1) What are the major topics and thematic areas investigated in clinical trials of DHIs for CRC screening? (2) What individual, contextual, technological, and timing-related factors influence CRC screening uptake across diverse populations in DHI-based trials? (3) How do digital interventions compare to conventional CRC screening strategies in terms of measured outcomes, patient engagement, and short- and long-term sustainability? Individual factors may include age, race, gender, and literacy level; contextual factors include healthcare access, setting (rural vs. urban), or insurance coverage; technological factors refer to the type and functionality of the digital intervention; and time-related factors involve timing and duration of intervention delivery.

## Methods

This scoping review aims to provide a comprehensive synthesis of clinical outcomes related to CRC screening uptake through digital interventions, adhering to the Preferred Reporting Items for Systematic Reviews and Meta-Analyses for Scoping Review (PRISMA-ScR) guidelines [21,22] resulted in Fig 1. We conducted our literature search across three major databases: PubMed, Google Scholar, and ClinicalTrials.gov for studies related to DHIs and CRC screening. ClinicalTrials.gov was used to identify completed or ongoing clinical trials involving DHIs for CRC screening. While this platform does not index full-text articles, we used trial identifiers to cross-reference published results via PubMed or Google Scholar where possible. The full search strategy, including Boolean operators and filters, is provided in S1 Appendix. The search was conducted between March 1st and April 20th, 2024, with no restrictions on geography. Only articles published in English were included. These databases were chosen based on their extensive coverage of biomedical literature, clinical trial data, and digital health innovations, making them well-suited for identifying relevant studies on CRC screening.

Our search strategy focused on three primary concepts: (1) CRC screening (e.g., "colorectal neoplasms/diagnosis," "colorectal neoplasm screening"), (2) digital interventions (e.g., "mobile health," "telehealth," "patient portals"), and (3) health literacy (e.g., "patient education," "digital literacy"). We included health literacy in our search strategy to capture studies where patient understanding and engagement with digital tools could influence CRC screening outcomes. Emerging evidence suggests that digital health literacy mediates the effectiveness of DHIs, especially in underserved or low-literacy populations. Therefore, health literacy was an intentional conceptual focus aligned with Research Question 2.

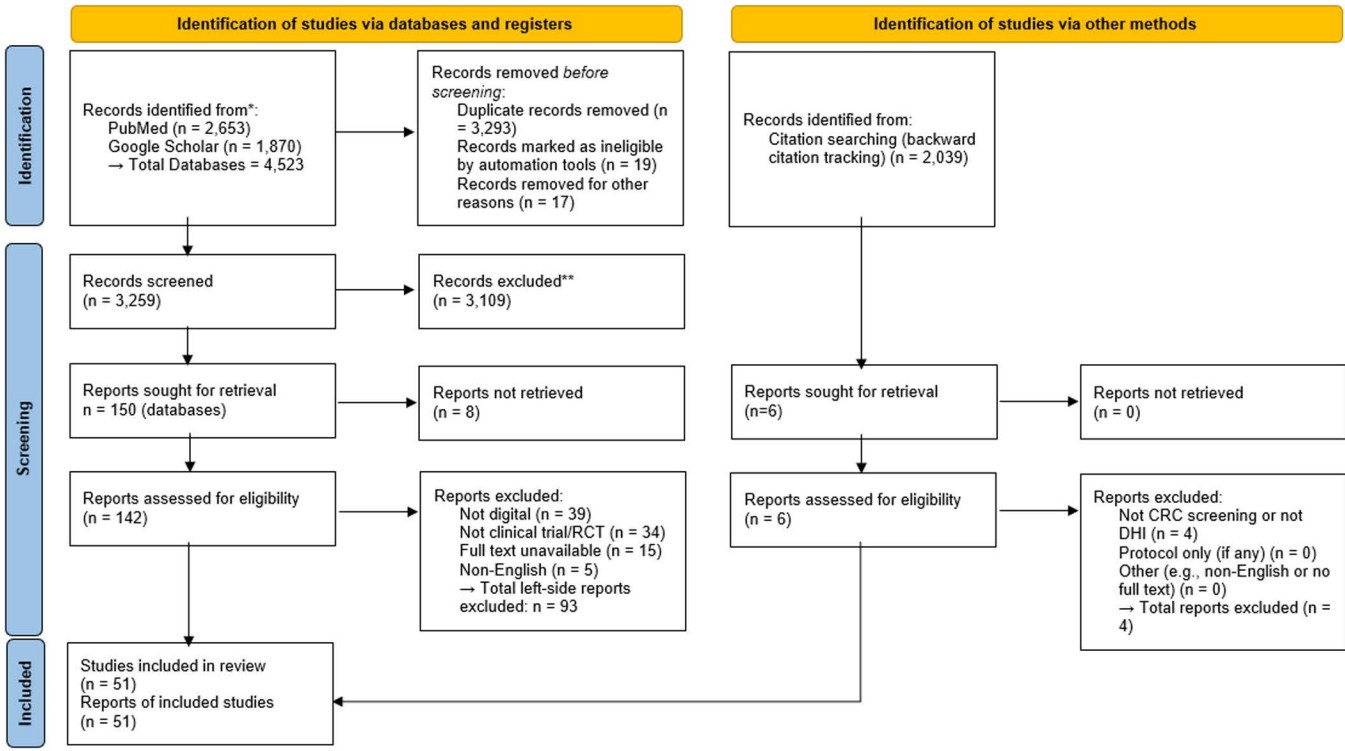

**Fig 1. PRISMA 2020 flow diagram of study selection.**

Boolean operators (AND, OR) were employed to combine these search terms effectively, with additional filters for clinical trials, randomized controlled trials, and studies published in English. A full description of the search strategy is available in S1 Appendix.

We limited inclusion to clinical trials or randomized controlled trials evaluating DHIs related to CRC screening. Studies were included if they focused on CRC screening in a broad sense and employed DHI such as mHealth apps, telehealth platforms, patient portals, or EMR systems. We included studies that targeted diverse populations across various healthcare settings, including primary care, community outreach, and hospital-based programs. Key concepts guiding inclusion were the use of technology to improve patient adherence to screening guidelines, enhance follow-up care, or facilitate patient education related to CRC screening.

We excluded studies that did not specifically focus on CRC screening or lacked a digital health intervention component. Additionally, studies that focused on non-human populations, non-English language publications, or interventions unrelated to healthcare delivery (e.g., lifestyle-only interventions) were excluded. Contextually, we considered studies conducted in various geographic regions, income levels, and healthcare systems to capture a broad range of interventions and outcomes. We excluded systematic reviews, meta-analyses, editorials, protocols, and observational studies unless they were embedded in a trial design. Data extraction followed the PICOT framework, which includes patient/population, intervention, control/comparison, outcome, and timing. Outcomes captured included both clinical outcomes (e.g., screening uptake, adherence) and nonclinical outcomes (e.g., knowledge, self-efficacy, satisfaction).

Two reviewers (SK & AW) independently screened the titles, abstracts, and full texts of the articles to ensure relevance and eligibility for inclusion in the scoping review. Any disagreements during the screening process were resolved through consensus discussions with a third reviewer (AB). To ensure comprehensive coverage, we systematically performed

backward citation tracking in Web of Science for 46 of the 49 included studies, identifying 2,039 cited references, which resulted in 1,266 unique records after deduplication. These were screened using the same inclusion and exclusion criteria. Six full-text reports were assessed for eligibility, and two studies met our criteria and were added to the final dataset. Forward citation tracking did not yield additional eligible studies (as of April 2024). These additions are reflected in the updated PRISMA 2020 flow diagram (Fig 1). Two reviewers (SK & AW) independently extracted data using a structured charting form aligned with PICOT domains to summarize key study characteristics. The data extraction process captured key variables, including participant demographics, type of digital intervention, CRC screening outcomes, study design, and intervention duration. Any discrepancies in data extraction were discussed and resolved by consensus with the third reviewer (AB) to ensure accuracy and consistency in the charting process.

To ensure transparency and alignment with our research aims, we explicitly structured data extraction and synthesis to correspond to each research question. For Research Question 1, titles and abstracts from the final 51 included PubMed records were used for bibliometric co-occurrence mapping in VOSviewer (version 1.6.20); this was combined with qualitative thematic synthesis to interpret clusters and group them into four conceptual domains. For Research Questions 2 and 3, we extracted detailed information from the full texts, including participant demographics, intervention type, comparators, and outcomes. These data informed the thematic factors summarized in Table 1 and the detailed study-level synthesis presented in S2 Appendix. Data from the included studies were charted based on key variables such as participant demographics, type of digital intervention, CRC screening outcomes, and intervention duration. The full list of data items charted from the included studies is detailed in S2 Appendix, to organize participant, intervention, and outcome data.

Since the objective of this scoping review was to map the available literature on DHIs for CRC screening and identify research gaps, no formal critical appraisal of the included studies was conducted. This approach is consistent with scoping review methodology, which aims to provide a comprehensive overview of the evidence base rather than assess the quality of individual studies. As a result, this review focuses on summarizing the scope and characteristics of the interventions without evaluating the methodological rigor or risk of bias.

"Keyword co-occurrence mapping was performed using VOSviewer (version 1.6.20), a validated tool for bibliometric network visualization [23,24] based on titles and abstracts from the final 51 PubMed records." To standardize terminology, a custom thesaurus file was applied to merge synonyms for key concepts. The analysis used full counting with association strength normalization and set a minimum occurrence threshold of six, identifying 199 terms grouped by the software into ten initial clusters.

For interpretive synthesis, these ten clusters were merged into four broader thematic domains: (1) Patient-Centered Technology & Adherence, (2) Behavioral Design & Personalization, (3) Clinical Workflow & Provider Interaction, and (4) Equity, Disparities & Community Engagement. Example keywords from each domain were documented to illustrate the rationale for grouping. This consolidation enhanced alignment with our PICOT framework and strengthened the linkage between Research Questions 1 and 2.

For thematic analysis (Research Question 1), keyword co-occurrence mapping relied solely on titles and abstracts. The VOSviewer settings used default parameters, including full counting and association strength normalization. A simple co-author network showed 337 unique authors with two main collaboration clusters; the visualization highlights regional subgroups aligning with known institutional hubs, reflecting strong intra-cluster ties and bridging connections across the broader CRC screening research network. As mentioned, for Research Questions 2 and 3, a full-text review was conducted by two independent reviewers who manually coded each study using the PICOT framework.

## Results

### Key topics and thematic domains in CRC screening research (RQ1)

Our search across PubMed, Google Scholar, and ClinicalTrials.gov identified 2,820 records (PubMed: 2,653; Google Scholar: 141; ClinicalTrials.gov: 26), as shown in Fig 1. After screening and deduplication, 51 studies were included in our

**Table 1. Key findings from PICOT analysis.**

| PICOT Component | Major Findings | Research Gaps |
|---|---|---|
| Population in Focus | • Focus on populations aged 50+based on CRC screening guidelines.<br>• Inclusive of minority populations and different demographics in urban and rural settings.<br>• Studies target gender-specific and ethnic community-specific factors influencing CRC screening. | • Under-50 population rarely targeted.<br>• Limited studies on non-Spanish speaking minorities and unique health exposures like veterans and immigrants.<br>• Recruitment strategies often miss broader community representation. |
| Digital Intervention Approach | • Shift towards patient-centered, digital health solutions like mHealth, apps, and patient portals.<br>• Use of diverse intervention methods such as personalized messages, navigation aids, and telephonic support.<br>• Integration with EMRs to enhance reminder systems and screening adherence. | • Reliance on traditional communication methods limits exploitation of advanced digital technologies.<br>• Insufficient measurement of digital and health literacy impacts.<br>• Lack of innovative intervention designs and detailed reporting on intervention delivery variability. |
| Comparison to Conventional Approach | • Most studies used conventional care as the comparator.<br>• Usual care typically included mailed reminders, in-clinic counseling, or standard referrals.<br>• Some studies added minimal digital elements (e.g., automated calls) to comparators to isolate intervention effects. | • Few studies compared outcomes by CRC stage (early vs. late-onset).<br>• Limited longitudinal follow-up to assess sustained effects.<br>• Lack of analysis on how digital interventions influence screening uptake over time. |
| Outcomes as Success Indicators | • Positive outcomes in CRC screening rates from tailored interventions and technology use.<br>• Detailed effectiveness and completion rates provided for screening modalities like FIT, FOBT, and colonoscopy.<br>• Outcomes discuss immediate to long-term impacts on screening behaviors and CRC detection. | • Need for more comprehensive studies addressing integration of digital interventions, patient compliance, and long-term impacts.<br>• Limited differentiation of interventions based on CRC risk levels and stages.<br>• Scarcity of research on follow-up screening practices, especially in digital contexts. |
| Temporal Understanding of Digital Intervention | • Timeframes for outcome measurement range from 6 months to several years, allowing evaluation of short-term efficacy and long-term sustainability.<br>• Specific timing of interventions sometimes tied to clinic visits or seasons, influencing engagement and effectiveness. | • Limited detail on timing components in studies, affecting the understanding of intervention impacts.<br>• Need for studies to differentiate outcomes based on CRC stages at intervention time.<br>• Essential to integrate comprehensive risk factors with timing for personalized screening schedules. |

Note: "Outcomes" include both clinical (e.g., screening uptake, adherence) and nonclinical measures (e.g., knowledge, intent, satisfaction) measures.

final PICOT analysis. The included studies (n = 51) analyzed in this scoping review are cited in-text using author–year format for clarity, but they are not included in the manuscript's formal reference list, as they were reviewed as data sources rather than cited literature. A bibliometric overview using Web of Science and PubMed confirmed that these studies were published between 2015 and 2024, primarily in high-impact clinical journals such as Annals of Internal Medicine and JAMA Internal Medicine, as well as interdisciplinary journals like American Journal of Public Health and Implementation Science Communications. Other publications appeared in journals on telemedicine and digital health, reflecting the diverse range of research addressing CRC screening through digital interventions.

The majority of the research focused on CRC screening through digital health tools, including mobile applications, patient portals, telehealth strategies, and reminder systems. Contributions from behavioral sciences, informatics, and implementation research add further depth to understanding how these interventions are designed and deployed across diverse populations. Geographic representation was dominated by the United States, which contributed over 50% of the included studies, but countries such as Australia, the Netherlands, Spain, and the United Kingdom also provided significant contributions. Many studies addressed local and regional disparities in CRC screening and emphasized how tailored digital approaches can improve screening uptake among underserved or high-risk groups. Study sites were heterogeneous, ranging from urban academic medical centers, integrated health systems, and Federally Qualified Health Centers

(FQHCs) in the United States, to rural community clinics and regional networks internationally (e.g., Australia, Malaysia, South Korea, and the Netherlands). Approximately 70% were conducted in US settings, but some focused on underrepresented groups such as Black, Hispanic/Latino, and American Indian communities. Full site details are provided in S2 Appendix.

Thematic analysis of co-occurring keywords using VOSviewer (version 1.6.20) was conducted to map core topical areas. After applying a custom thesaurus and setting a minimum occurrence threshold of six, 199 terms were included in the final network, grouped by VOSviewer into 10 clusters based on co-occurrence strength. For reporting clarity, these clusters were then conceptually synthesized into four overarching thematic domains: (1) Patient-Centered Technology & Adherence — covering core CRC screening delivery, digital reminders, and adherence support (e.g., "mHealth", "app", "SMS", "patient portal", "participation"); (2) Behavioral Design & Personalization — focusing on digital health literacy, usability, and personalized content (e.g., "digital health literacy", "acceptability", "self-efficacy"); (3) Equity, Disparities & Community Engagement — highlighting terms linked to culturally tailored approaches (e.g., "minority groups", "Black woman", "Black man", "race matching", "rural"); and (4) Clinical Workflow & Implementation Outcomes — reflecting keywords related to practical delivery, adherence, cost-effectiveness, and sustainability (e.g., "adherence", "decision support", "quality of life", "navigation").

In Fig 2, the centrality of terms such as "cancer screening", "CRC screening uptake", and "participation" reflects a research emphasis on patient engagement through digital tools. Keywords like "app", "mHealth", "SMS", and "patient portal" reflect the significant role of mobile and web-based platforms in facilitating education, reminders, and behavior change. Studies using text messaging or app-based interventions for reminders and follow-up (e.g., Chan, 2008; Elepaño, 2021) illustrate this trend.

In addition to technological delivery, a clear cluster of terms such as "digital health literacy," "acceptability," and "self-efficacy" highlights how patient understanding and usability considerations shape engagement and adoption. Studies addressing health literacy, digital skills, and personalization indicate the need for adaptive design and support systems.

Another dominant area is equity and contextual factors, shown by terms like "minority groups," "Black woman," "Black man," "race matching," and "rural." These keywords reflect an emphasis on culturally tailored interventions to reduce screening disparities. Several studies specifically targeted African American, Hispanic/Latino, or rural populations through bilingual or community-embedded approaches (Coronado, 2023; Lohr, 2023; Wilson-Howard, 2021).

Finally, the map indicates the importance of practical implementation factors and outcomes, with nodes related to "adherence," "quality of life," "participation rate," and "cost-benefit analysis." These link the research to sustainability and long-term impact considerations. Studies that included decision support tools, patient navigation, or multicomponent interventions show the diverse strategies used to improve follow-up and completion rates (Basch, 2006; Gomez, 2023).

The characteristics of the included studies, including population demographics, intervention designs, and outcome measures, are summarized in Table 1 and detailed in S2 Appendix. Overall, the evolving landscape of CRC screening research reflects the increasing adoption of digital tools, a shift toward patient-centered and personalized care, and an urgent need to address health equity and long-term effectiveness. These themes directly inform the classification of interventions and delivery strategies discussed in Research Question 2.

### Individual, contextual, technological, and temporal factors influencing screening uptake (RQ2)

Our PICOT analysis identified key demographic, socioeconomic, and digital intervention-related factors that impact CRC screening uptake. Age was a common focus, with 12 studies targeting populations aged 50 and above, particularly the 50–75 age group, while 47 studies either covered broader or unspecified age ranges. Specific studies, such as Hong, et al. (2014) focused on narrower age groups like 51–58, aligning with screening guidelines. In terms of gender, one study exclusively focused on male populations, while four focused on female populations, particularly breast and CRC screenings (Champion, 2020; Champion, 2023; Vilaro, 2020). The remaining studies either involved mixed-gender groups or did not specify gender. Table 1 highlights key demographic findings.

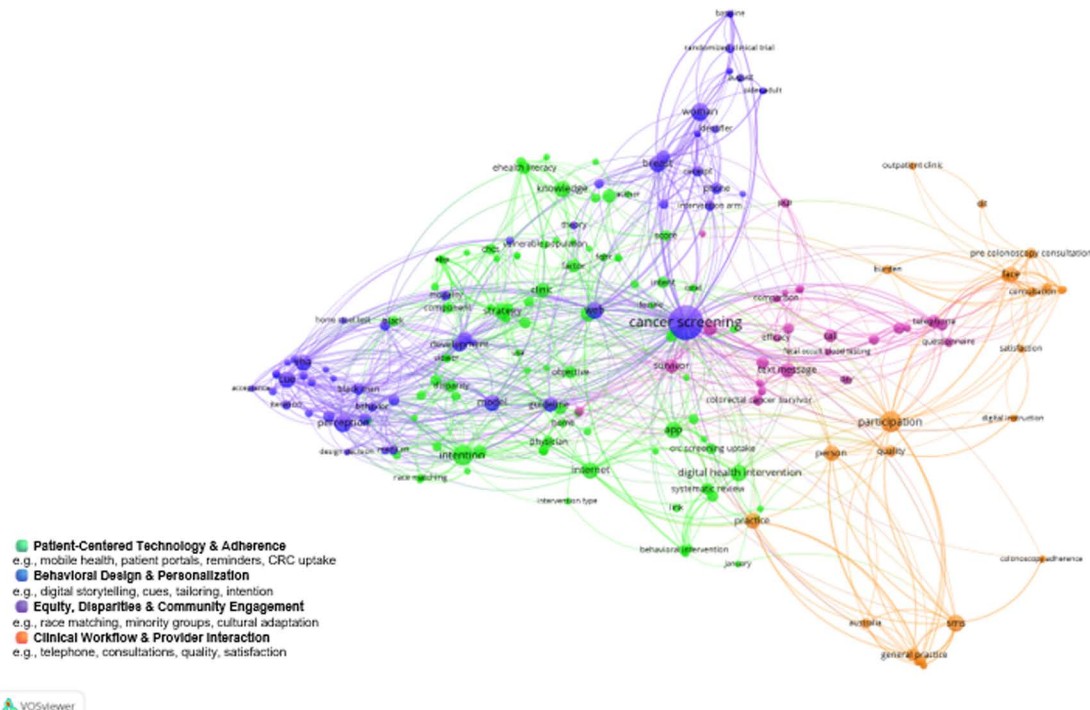

**Note**: Co-occurrence network map of keywords from the included 51 CRC screening studies, generated using VOSviewer (version 1.6.20). Nodes represent keywords (titles and abstracts) merged using a custom thesaurus; node size indicates the frequency of occurrence, and link thickness reflects the strength of co-occurrence relationships (association strength normalization, full counting). Colors indicate software-generated clusters grouped into four conceptual domains; node size represents the relative frequency of term co-occurrence.

**Fig 2. Keyword co-occurrence network for included studies, grouped by VOSviewer clustering and synthesized into four major themes.**

Additionally, minority populations, including urban minorities, rural communities, and ethnic groups like Latinos and Hispanic/Latinos, were studied in 24 articles. Hispanic populations in Southern California were a focal point (Coronado, 2023; Lohr, 2023; Wilson-Howard, 2021), while Black communities in New York and North Florida were highlighted in others (Basch, 2006; Cooks, 2022; Vilaro, 2020; Vilaro, 2021; Wilson-Howard, 2021). Rural populations faced specific healthcare access challenges, with several studies focusing on regions like Appalachia (Baus, 2020) and states like Kentucky and West Virginia (Baus, 2020). The distinction between rural and urban healthcare access barriers was explored in various regions, including North Carolina, Texas, California, and New York (Cohen-Cline, 2014; Gautom, 2023; Halm, 2023). Some interventions were embedded in community settings, leveraging churches and health centers to improve CRC screening uptake (Gomez, 2023). Complete review results by individual articles is available at S2 Appendix.

Our analysis showed a significant shift towards technology-enhanced, patient-centered interventions to improve CRC screening adherence. Various digital interventions were utilized, including personalized electronic messages, tailored telephone outreach, video decision aids, mHealth tools, and telephonic patient navigation. As detailed in Table 2, DHIs were categorized into three primary delivery modes: technology-only (e.g., web portals, SMS, apps), human-facilitated (e.g., patient navigation, counseling, live calls), and hybrid models (e.g., multimedia tools paired with follow-up navigation). Tools such as patient portals, apps, and automated outreach systems were commonly employed in technology-only models (Baus, 2020; Green, 2013; Halm, 2023; Lafata, 2019), while hybrid approaches frequently combined digital content with tailored human interaction (Champion, 2020; Champion, 2023; Rawl, 2024). Human-facilitated interventions centered on direct phone calls or counseling strategies (Basch, 2006; López-Torres Hidalgo, 2016; Selva, 2019; Wu, 2019).

**Table 2. Classification of Included Studies by Digital Delivery Mode and Human Facilitation.**

| Delivery Mode | Intervention Type | Studies (First Author, Year) |
|---|---|---|
| Technology-only | EHR/Portal/Online Tools | Baus 2020; Chan 2008; Goshgarian 2022; Green 2013; Halm 2023; Lafata 2019; Misra 2011; Richter 2020; Saini 2023; Sequist 2011 |
| | SMS/Text Messaging | Henderson 2022; Hirst 2017; McIntosh 2023; McIntosh 2024; Mosen 2010; Muller 2017; Schliemann 2022; Van Blarigan 2020; Wu 2019 |
| | Virtual Assistants/Agents | Cooks 2022; Vilaro 2020; Vilaro 2021; Vilaro 2022; Wilson-Howard 2021; Zalake 2019; |
| | IVR/Automated Calls | Cohen-Cline 2014 |
| | mHealth/Tablet Decision Tools | Denizard-Thompson 2020; Miller 2018; Wyse 2023 |
| | Risk Assessment | Yen 2021 |
| | Meta-Analysis | Elepaño 2021 |
| | eHealth Literacy/Survey | Mitsutake 2012 |
| Hybrid | Telephone-based & Navigation | Basch 2006; Hong 2014; López-Torres Hidalgo 2016; Scott 2023; Selva 2019; Menon 2022; Stoop 2012 |
| | Multimedia + Human Support | Champion 2020; Champion 2023; Dodd, 2017; Gomez 2023; Lohr 2023; McQueen 2019; Rawl 2024 |
| | Home FIT Kits + Digital Support | Coronado 2023; Jerant 2015; Malo 2021; Schliemann 2022 |
| | Development/Community Engagement | Gautom 2023 |
| | Virtual Agents with Navigation | Vilaro 2020; Cooks 2022; Vilaro 2021; Vilaro 2022; Wilson-Howard 2021; Zalake 2019 |
| Human-facilitated | Telephone-based Outreach | Basch 2006; Hong 2014; López-Torres Hidalgo 2016; Selva 2019; Scott 2023; Stoop 2012 |
| | Patient Navigation Only | Champion 2023; Menon 2022 |

**Note:** Each of the 51 studies is categorized by its primary intervention delivery mode: Technology-only, Hybrid (combined digital and human support), or Human-facilitated (e.g., phone outreach or navigation with minimal digital components). Full details, including study sites and PICOT elements, are provided in S2 Appendix.

Multilevel interventions tackled barriers at patient, provider, and system levels through combinations of education, reminders, and navigation support (Halm, 2023; Malo, 2021; Saini, 2023; Schliemann, 2022). Some interventions integrated with EMR to automate reminders and enhance screening adherence (Goshgarian, 2022; Green, 2013; Halm, 2023). Mobile apps and web-based tools were particularly effective in delivering reminders and decision-making aids (Elepaño, 2021; Wyse, 2023). Additionally, telecommunication tools like interactive voice response systems and telephone outreach also showed promise (Basch, 2006; Cohen-Cline, 2014; Gomez, 2023; Stoop, 2012). Screening modalities like mailed FIT, often paired with follow-ups and reminders, were commonly used to improve screening rates (Coronado, 2023; Malo, 2021; Scott, 2023). Studies also focused on improving colonoscopy adherence using decision aids, navigators, and pre-procedure digital tools (Schliemann, 2022; Stoop, 2012).

### comparative effectiveness and time-based outcomes of digital vs. conventional screening (RQ3)

Most CRC screening studies compared digital interventions with conventional care, such as mailed reminders or routine care, which served as benchmarks to evaluate the added value of digital strategies. Some studies used pre-visit educational content or prior assessments (Denizard-Thompson, 2020; Schliemann, 2022; Scott, 2023), while others focused on colonoscopy interventions, using traditional pre-procedure consultations as comparators. Occasionally, control groups received basic electronic messages or reminders to measure the effectiveness of more robust interventions (Hirst, 2017; Sequist, 2011).

Outcomes from digital interventions were mostly positive. For example, tailored telephone outreach increased screening rates by 20.9% (Basch, 2006), and data-driven strategies improved rates from 28.4% to 49.5% (Baus, 2020). Interventions using interactive voice response systems and mHealth tools resulted in a 2.2% rise in screening rates (Cohen-Cline, 2014; Elepaño, 2021). The combination of mailed fecal tests with navigation support significantly boosted adherence, with rates increasing from 26.3% to 64.7% over two years (Coronado, 2023; Green, 2013). However, some interventions showed limited impact. For instance, FOBT return rates showed minimal improvement in certain studies (Chan, 2008). Others focused more on screening intentions than outcomes (Cooks, 2022), and biweekly SMS reminders showed only modest gains in colonoscopy adherence (Wu, 2019). Telephone consultations were found to be less effective than in-person consultations (Stoop, 2012). These terms were connected in the VOSviewer map and also reexamined through full-text synthesis. Key measures included screening uptake rates, completion rates, and patient-provider communication, with digital interventions generally outperforming conventional care. Different screening modalities, such as FIT, FOBT, and colonoscopy, were evaluated in various populations, with timeframes ranging from short-term (6 weeks) to long-term (several years), impacting screening behavior and CRC detection. The timeframe for measuring outcomes varied significantly, typically ranging from 6 months to several years, allowing for evaluations of both short-term efficacy and long-term sustainability. Some studies had shorter durations of 6 weeks (Dodd, 2017), while others extended up to 10 years (Halm, 2023) with most studies lasting 6 months to 2 years, which is crucial for assessing medium-term outcomes.

Timing and duration were influenced by the study design and intervention type. Some interventions were strategically timed before clinic visits or during specific seasons, which affected participant engagement due to health behaviors and access to healthcare facilities (Rawl, 2024; Champion, 2023). Exact months or years were specified in several studies to contextualize outcomes relative to external factors such as pandemic-related service disruptions or public awareness campaigns. While most interventions reported implementation duration retrospectively, ongoing studies like Rawl et al. (2024) incorporated multi-year follow-up to capture long-term behavioral impact. Overall, study durations reflect diverse research goals, from short-term efficacy to sustained screening adherence, underscoring the need for continued evaluation of digital interventions' long-term effects on CRC screening behaviors.

## Discussions

Our extensive review of literature on CRC screening and digital interventions highlights significant advances while delineating areas requiring further investigation. This discussion synthesizes major findings from the PICOT analysis and proposes directions for future research. First, our PICOT analysis reveals that most CRC screening studies focus on populations aged 50 and above, with some addressing narrower age ranges and exploring demographic impacts on screening behavior among gender-diverse and ethnic minority groups, including specific urban and rural communities in detailed settings. However, an important consideration is the recent change in U.S. guidelines, which lowered the recommended screening age from 50 to 45 in May 2021 [25] due to the rising incidence of early-onset CRC. Approximately 35 of the studies included in our review were published before this change, and they reflect older guidelines. Among studies that reported age data, few explicitly focused on adults under 50, limiting our ability to assess how well early-onset CRC populations are represented in digital health interventions.This site-level detail confirms that our included studies reflect diverse healthcare contexts and screening infrastructures, which is important when interpreting variation in outcomes and generalizability. Moreover, some non-US settings may have differing guidelines for CRC screening age or test modality, which adds to the heterogeneity. Moreover, some studies were conducted outside the U.S., where different screening age recommendations may apply. These factors highlight a need for caution when interpreting results across different regions and time frames.

Given the shifting epidemiology of CRC, urgent research is needed to evaluate the adequacy of screening protocols, particularly for younger populations under the new guidelines. This includes exploring the utility of digital interventions and other adjuncts in enhancing CRC screening uptake, which can play a pivotal role in improving both prevention and

early detection. Furthermore, the scarcity of data on CRC screening uptake in high-risk populations, who frequently begin screening prior to the recommended age of 45, presents a critical gap in need of attention. Addressing these research gaps is essential to ensure that CRC prevention strategies evolve alongside current epidemiological trends and risk profiles. Moreover, while there is commendable focus on diverse demographic groups including minorities and rural populations, specific vulnerable groups such as veterans, immigrants, and non-Spanish speaking minorities are less represented. Future studies should extend their demographic inclusivity, enhancing screening strategies tailored to the unique health profiles and access challenges of these populations. This could involve deploying community-based interventions that leverage local structures for wider reach and impact.

Secondly, the shift towards DHIs marks a progressive step in patient-centered care, employing tools from telephonic navigation to sophisticated mobile health apps and patient portals. Despite these advancements, much of the research still relies on traditional communication methods such as phone calls and text messages, which may not fully utilize the capabilities of modern digital technology like real-time analytics and personalized patient engagement strategies. For example, an email intervention may not be effective if the population being targeted does not have reliable internet access, highlighting the need for careful consideration of accessibility in the design of these interventions. Additionally, there is a conspicuous lack of studies addressing digital and health literacy, which are pivotal in determining how effectively patients can engage with and benefit from digital health tools. Despite increasing reliance on digital modalities such as web-based modules and virtual health assistants [15,17], most interventions did not assess digital health literacy (DHL) or adapt their design accordingly. Future CRC screening trials should integrate DHL screening tools to tailor interventions by digital proficiency and health engagement levels. Research expanding on digital literacy would provide deeper insights into patient interactions with digital systems, potentially guiding the development of more intuitive and accessible digital health platforms.

Thirdly, the prevalent use of conventional care as comparators underscores a persistent reliance on established screening methods. This reliance might be limiting the exploration of innovative, potentially more effective DHIs. Comparative studies focusing on different stages of CRC, particularly distinguishing between early and late-onset CRC, are scant. Such comparative research could elucidate differential impacts of screening interventions across various CRC stages, informing stage-specific screening strategies that might enhance prognosis and treatment outcomes.

Fourth, our review predominantly shows positive results, demonstrating the effectiveness of various approaches such as tailored telephone outreach, personalized communication strategies, and mHealth interventions, which have significantly shifted screening behaviors and improved outcomes. However, some interventions report only minimal improvements or lack detailed effectiveness data, highlighting the variability in intervention success and the nuanced impacts on healthcare systems. Additionally, there is a significant gap in longitudinal studies that monitor the long-term effects of CRC screening interventions. Such long-term data are essential for assessing the sustainability and real-world efficacy of these interventions, especially in understanding how digital interventions perform over extended periods and under various patient health conditions. Moreover, while outcomes like screening uptake rates and patient knowledge are commonly reported, there is a need for more comprehensive outcome measures that include long-term health impacts, quality of life, and cost-effectiveness. Although most studies reported screening uptake as the primary endpoint, other important outcomes such as patient satisfaction, cost-effectiveness, long-term adherence, and equity impacts were rarely explored. These gaps highlight the need for future studies to adopt comprehensive outcome frameworks aligned with the RE-AIM (Reach, Effectiveness, Adoption, Implementation, and Maintenance) model. Specifically, digital literacy related to CRC screening could identify mitigating factors that influence the effectiveness of long-term interventions. Longitudinal studies should also aim to integrate diverse factors relevant to clinical outcomes, such as the effects of comorbid conditions, which can significantly impact CRC risk and screening efficacy.

Lastly, our findings identified that the timeframe for measuring outcomes in CRC screening digital interventions varies from 6 months to several years, essential for assessing both short-term efficacy and long-term sustainability. Details are

often limited, with study durations ranging from 6 weeks to 10 years, primarily focusing on 6 months to 2 years. Specific timing of interventions can significantly influence outcome effectiveness, with exact dates crucial for understanding the impacts of contextual and environmental factors on study results. The current literature lacks a detailed exploration of how the timing of digital communications and interventions influences patient compliance and screening uptake. Future research should focus on the timing and frequency of digital interventions to maximize their impact, considering patient behavior and engagement patterns.

Our study has several limitations, particularly in capturing the most recent advancements in DHIs due to the nature of clinical trial research. A key limitation is the exclusion of usability testing, which is often not included in the clinical trial data available through PubMed. PubMed is a predominant source for clinical literature, including many technology-driven interventions; however, it may not fully capture the latest AI-driven advancements and usability testing studies. Clinical trials tend to prioritize efficacy and safety, while aspects such as user experience and usability from system design and testing are often overlooked. As a result, digital tools tested in clinical settings may not be fully optimized for user engagement or effectiveness in real-world applications. Additionally, the rapid development of technologies, especially in areas like artificial intelligence, can outpace the traditional clinical trial process. Many cutting-edge technologies are assessed in preliminary feasibility studies rather than full-scale interventional trials, limiting the available evidence on their long-term efficacy and practical utility. The integration of new technologies into clinical trials is often delayed by lengthy regulatory and ethical review processes, which can further slowdown the assessment of innovative DHIs.

Moreover, the lack of formal critical appraisal of study quality may limit the interpretation of results in this scoping review. As the objective was to map the available literature, we did not assess the methodological rigor or risk of bias in the included studies. This variability in study quality might affect the reliability of the reported outcomes across studies. The availability of data also varied significantly, with many studies lacking detailed reporting on long-term outcomes and screening adherence over time, limiting the ability to evaluate the sustainability of digital interventions. Finally, some of the included studies had small sample sizes or were conducted in specific geographic or healthcare settings, affecting the generalizability of findings. The heterogeneity in digital interventions, study designs, and populations further complicates direct comparisons between studies. These limitations highlight the need for more robust, high-quality research to better understand the long-term impact and effectiveness of DHIs in diverse populations.

## Conclusion

Our comprehensive review of literature on CRC screening interventions highlights the evolving landscape of screening strategies, where technology and personalized approaches play pivotal roles. Digital tools such as mobile health applications, web platforms, and interactive voice responses are proving instrumental in enhancing patient engagement and screening adherence. Despite the success of many interventions, challenges persist in reaching certain populations and ensuring consistent outcomes across diverse groups. The integration of patient-centric practices within clinical protocols is crucial for improving the effectiveness of screening programs. While the body of research on CRC screening is robust, addressing these identified gaps could significantly enhance screening strategies, making them more personalized, timely, and effective.

## Supporting information

**S1 Appendix. Search queries.**
(DOCX)

**S2 Appendix. PICOT analysis.** Summarizes the PICOT (Population, Intervention, Comparator, Outcome, Time) framework analysis of 51 included studies. Presents structured tabular data of each study's design, population, and key outcomes, along with full citations of all reviewed articles. This appendix supports the thematic synthesis presented in the Results section.
(DOCX)

**S1 Checklist. PRISMA checklist.**
(DOCX)

## Author contributions

**Conceptualization:** Sujin Kim, Jihye Bae, Avinash Bhakta.

**Data curation:** Sujin Kim.

**Formal analysis:** Sujin Kim, Andrew J. Whipkey.

**Funding acquisition:** Sujin Kim.

**Investigation:** Andrew J. Whipkey.

**Methodology:** Sujin Kim.

**Project administration:** Sujin Kim.

**Supervision:** Avinash Bhakta.

**Validation:** Andrew J. Whipkey.

**Visualization:** Sujin Kim.

**Writing – original draft:** Sujin Kim.

**Writing – review & editing:** Sujin Kim, Andrew J. Whipkey, Jihye Bae, Avinash Bhakta.

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
