## [Decision Letter · Decision Letter 0]

20 Apr 2025

Response to Reviewers
Revised Manuscript with Track Changes
Manuscript
**Journal Requirements:**
**Additional Editor Comments (if provided):**
**Reviewers' Comments:**

**Comments to the Author**

1. Does this manuscript meet PLOS Digital Health’s publication criteria?

Reviewer #1: Partly

Reviewer #2: Partly

2. Has the statistical analysis been performed appropriately and rigorously?

Reviewer #1: Yes

Reviewer #2: N/A

3. Have the authors made all data underlying the findings in their manuscript fully available (please refer to the Data Availability Statement at the start of the manuscript PDF file)?

Reviewer #1: No

Reviewer #2: Yes

4. Is the manuscript presented in an intelligible fashion and written in standard English?

Reviewer #1: Yes

Reviewer #2: Yes

Reviewer #1: Recommendation:

- Include a section evaluating the methodological rigor of the studies, using tools such as the Joanna Briggs Institute’s Critical Appraisal Tools.

- The variability in study durations (6 weeks to 10 years) and lack of standardized outcome measures hinder comparability. Discuss how these variations influence the interpretation of results and consider suggesting standard metrics for future studies.

-Add a discussion on the role of digital and health literacy, supported by relevant studies, to provide a holistic understanding of intervention effectiveness.

-The manuscript primarily reports screening uptake rates without delving into other critical metrics such as patient satisfaction, cost-effectiveness, or long-term health impacts. Broaden the scope of outcome measures to provide a more comprehensive evaluation of intervention success.

Reviewer #2: The manuscript, titled “ Digital Health Interventions for Colorectal Cancer Screening Update: A Scoping Review,” examined the effectiveness of digital health interventions (DHIs) applied to enhance colorectal cancer screening. The authors conducted a scope review guided by the PICOT framework and systematically searched three databases (i.e., PubMed, Google Scholar, & ClinialTrials.gov) for relevant articles. Through two reviewers’ independent study selection, data extraction, and analysis, the authors identified 58 relevant studies. The major findings support the effectiveness of DHI interventions but also disclose a notable gap in research targeting adults under 50. The review offers useful insights for guiding future research on DHIs and colorectal cancer screening outcomes.

Major comments:

1. While the authors stated the specific research aims, explicit research questions are not listed in the introduction. Since the results section was organized by research question, it is recommended that the authors clearly list the three research questions upfront.

2. In the methods section, the inclusion of “health literacy” in the search strategy should be justified in relation to the study’s aims or research questions. If this concept was intentionally included, one or more research questions should define the concept and reflect this focus.

3. The search strategy appears to focus on articles reporting on clinical trials involving DHIs. Therefore, the exclusion criteria should clearly state that non-clinical trial articles were excluded. In addition, it is unclear whether publication types (e.g., reviews, perspective articles) were part of the eligibility criteria. For example, Brenner (2014), cited in the results section (Table 2) as one of the included studies, is a systematic review and meta-analysis of randomized controlled trials and observational studies, which may conflict with the stated inclusion criteria.

4. The detailed search strategy was provided for PubMed as an appendix. ClinicalTrials.gov, unlike PubMed, is a registry database of clinical studies, not a bibliographic database. It does not contain journal articles directly. So, the article search from ClinicalTrials.gov needs to be explained. It is recommended to present the full search strategies for all three sources in the appendix.

5. The methods section should describe how the extracted data were analyzed and synthesized in relation to the research questions. For example, using a bibliometric approach to disclose the topic distributions. The bibliometric tools (e.g., Web of Science and VOSviewer should be explained in the methods.

6. The biometric results raise a couple of issues: (1) The geographic data from Web of Science – does it represent the location of study authors/coauthors or the clinical trial sites? (2) The co-occurrence keyword map generated using VOSviewer should include the number of keywords analyzed or visualized.

7. Article IDs used in the results should be replaced with appropriate in-text citations referring to the included articles.

**Do you want your identity to be public for this peer review?** For information about this choice, including consent withdrawal, please see our Privacy Policy

Reviewer #1: No

Reviewer #2: No

**Figure resubmission:****Reproducibility:** To enhance the reproducibility of your results, we recommend that authors of applicable studies deposit laboratory protocols in protocols.io, where a protocol can be assigned its own identifier (DOI) such that it can be cited independently in the future. Additionally, PLOS ONE offers an option to publish peer-reviewed clinical study protocols. Read more information on sharing protocols at https://plos.org/protocols?utm_medium=editorial-email&utm_source=authorletters&utm_campaign=protocols

---

## [Decision Letter · Decision Letter 1]

9 Jun 2025

Response to Reviewers
Revised Manuscript with Track Changes
Manuscript
**Journal Requirements:**
**Additional Editor Comments (if provided):**
**Reviewers' Comments:**

**Comments to the Author**

Reviewer #1: All comments have been addressed

Reviewer #2: All comments have been addressed

publication criteria?

Reviewer #1: Yes

Reviewer #2: Partly

3. Has the statistical analysis been performed appropriately and rigorously?

Reviewer #1: Yes

Reviewer #2: N/A

4. Have the authors made all data underlying the findings in their manuscript fully available (please refer to the Data Availability Statement at the start of the manuscript PDF file)?

Reviewer #1: Yes

Reviewer #2: Yes

5. Is the manuscript presented in an intelligible fashion and written in standard English?

Reviewer #1: Yes

Reviewer #2: Yes

Reviewer #1: Recommendations:

-Include younger and high-risk populations in future analyses.

-Incorporate digital and health literacy assessments into study selection and analysis.

-Apply formal quality appraisal tools to assess study reliability.

-Encourage studies using innovative technologies like AI and app-based engagement.

-Focus more on long-term outcomes and sustainability of DHIs.

-Enhance comparative analyses, particularly by CRC stage and intervention timing.

Reviewer #2: Please see the attachment.

**Do you want your identity to be public for this peer review?** For information about this choice, including consent withdrawal, please see our Privacy Policy

Reviewer #1: No

Reviewer #2: No

**Figure resubmission:****Reproducibility:** To enhance the reproducibility of your results, we recommend that authors of applicable studies deposit laboratory protocols in protocols.io, where a protocol can be assigned its own identifier (DOI) such that it can be cited independently in the future. Additionally, PLOS ONE offers an option to publish peer-reviewed clinical study protocols. Read more information on sharing protocols at https://plos.org/protocols?utm_medium=editorial-email&utm_source=authorletters&utm_campaign=protocols

---

## [Decision Letter · Decision Letter 2]

29 Aug 2025

Response to Reviewers
Revised Manuscript with Track Changes
Manuscript
**Journal Requirements:**
**Additional Editor Comments (if provided):**

Reviewer #2:

he authors thoughtfully addressed the reviewers’ comments from the previous two rounds and made significant adjustments to the manuscripts by adding more details about their methods and visualizations. These changes improved the transparency and readability. To further strengthen the manuscript and ensure it meets PLOS publishing standards, I offer the following suggestions:

1. Please refine the writing by reducing repetitive statements. For example, the PICOT framework is mentioned in Lines 134-135 and repeated with similar wording in Line 147. This also occurs elsewhere in the manuscript. After introducing an acronym like PICOT, maintain consistent use throughout. In addition, the title of Figure 1 includes search statistics, which were already displayed within the figure. To avoid redundancy, consider removing these duplicated descriptions.

2. Research Question 2 is about “individual, contextual, technological, and time-related factors influence CRC screening uptake.” However, the introduction did not cite sufficient existing literature explaining what these factors are and why they matter. For example, what individual factors refer to? What are typical contextual factors disclosed in previous literature? The meaning of “time-related factors?”

3. Research Questions 3 compares the digital health intervention to conventional CRC screening strategies. It would be helpful to cite previous studies and give examples of what constitutes conventional approaches. Table 1 lists the major findings related to “comparison to conventional approach,” but still does not illustrate what those conventional methods are. Providing context from previous studies (e.g., what is typically meant by “usual care”) would enhance clarity.

4. Line 95, “This scoping review aims to provide a comprehensive synthesis of clinical outcomes related to CRC screening uptake through …” It remains unclear what is meant by “outcomes” in this context. Does "outcome" refer to clinical outcome only in this review? Or does it include nonclinical outcomes as well? The aim appears to focus solely on clinical outcomes. However, Appendix B shows several nonclinical outcomes under the “outcome” column, such as “Development and refinement of culturally relevant patient educaiton materials for CRC screening” (Gautom, 2023), “CRC screening knowledge and self-efficacy” (Jerant, 2015), “patient -percieved benefits, barriers, and intent “ (Lafata, 2019). This manuscript should clearly define what consitutes an “outcome” in the context of the PICOT analysis.

5. The authors noted that this is a scoping review; no quality assessment was conducted. However, the stated aim (Line 84-85) reads “… this scoping review seeks to synthesize current evidence on the effectiveness of DHIs in increasing CRC screening rates, …” which may overreach given the absence of methodological quality assessment. I would suggest narrowing the stated aim to better align with the scoping review methodology standards.

6. The authors added more details about using VOSViewer settings to disclose topical areas. The authors should cite relevant literature that validates this method for topic mapping. The statements, such as (Line 231-232) “.. underscore the importance of engaging patients through digital tools,” may overinterpret the results. The co-occurrence mapping reflects research activity and trends, not causal relationships or impact. Consider rephrasing these interpretations with caution and accuracy.

**Reviewers' Comments:**

**Comments to the Author**

Reviewer #1: All comments have been addressed

Reviewer #2: All comments have been addressed

publication criteria?

Reviewer #1: Yes

Reviewer #2: Partly

3. Has the statistical analysis been performed appropriately and rigorously?

Reviewer #1: Yes

Reviewer #2: N/A

4. Have the authors made all data underlying the findings in their manuscript fully available (please refer to the Data Availability Statement at the start of the manuscript PDF file)?

Reviewer #1: Yes

Reviewer #2: Yes

5. Is the manuscript presented in an intelligible fashion and written in standard English?

Reviewer #1: Yes

Reviewer #2: Yes

Reviewer #1: i have no additional comments to author

all comments have been addressed

thank you very much

Reviewer #2: (No Response)

**Do you want your identity to be public for this peer review?** For information about this choice, including consent withdrawal, please see our Privacy Policy

Reviewer #1: No

Reviewer #2: None

**Figure resubmission:**

**Reproducibility:** To enhance the reproducibility of your results, we recommend that authors of applicable studies deposit laboratory protocols in protocols.io, where a protocol can be assigned its own identifier (DOI) such that it can be cited independently in the future. Additionally, PLOS ONE offers an option to publish peer-reviewed clinical study protocols. Read more information on sharing protocols at https://plos.org/protocols?utm_medium=editorial-email&utm_source=authorletters&utm_campaign=protocols

---

## [Editor Report · Decision Letter 3]

10 Sep 2025

Digital Health Interventions for Colorectal Cancer Screening Uptake: A Scoping Review

PDIG-D-24-00417R3

Dear Dr. Kim,

We're pleased to inform you that your manuscript has been judged scientifically suitable for publication and will be formally accepted for publication once it meets all outstanding technical requirements.

Within one week, you'll receive an e-mail detailing the required amendments. When these have been addressed, you'll receive a formal acceptance letter and your manuscript will be scheduled for publication.

An invoice for payment will follow shortly after the formal acceptance. To ensure an efficient process, please log into Editorial Manager at https://www.editorialmanager.com/pdig/ click the 'Update My Information' link at the top of the page, and double check that your user information is up-to-date. For billing related questions, please contact billing support at https://plos.my.site.com/s/.

Kind regards,

Dukyong Yoon

Section Editor

PLOS Digital Health